# Epidemiological Investigation of Yak (*Bos grunniens*) Fascioliasis in the Pastoral Area of Qinghai–Tibet Plateau, China

**DOI:** 10.3390/ani13213330

**Published:** 2023-10-26

**Authors:** Qijian Cai, Mengtong Lei, Chunhua Li, Jinzhong Cai, Doudou Ma, Houshuang Zhang

**Affiliations:** 1Key Laboratory of Animal Parasitology, Ministry of Agriculture and Rural Affairs, Shanghai Veterinary Research Institute, Chinese Academy of Agricultural Sciences, Shanghai 200241, China; xuanyun_cai@sina.com; 2Qinghai Academy of Animal Science and Veterinary Sciences, Xining 810016, China; lchxn@163.com (C.L.); caijinz@sina.com (J.C.); mdd092625@163.com (D.M.)

**Keywords:** *Fasciola* spp., epidemiology, investigation, yak, Qinghai–Tibet Plateau pastoral area

## Abstract

**Simple Summary:**

Infection of humans and animals with *Fasciola* spp. poses a threat to both the health of humans and animals and the animal industry. In the pastoral areas of the Qinghai–Tibet Plateau, yak is the most important economic animal, especially for the herdsmen and their animal products. However, the infection of yaks with *Fasciola* spp. is often underestimated. In this study, we investigated a total of 1784 yaks in 16 administrative counties in the Qinghai–Tibet Plateau region by fecal examination and autopsy of yaks after slaughter, and the differences in infection rates between different age groups and regions were analyzed. The clustering characteristics of the infection rates in spatial statistics were summarized. This investigation revealed that yaks on the Qinghai–Tibet Plateau had a high *Fasciola* spp. prevalence (17.32%), age was a risk factor for *Fasciola* spp. infection, and regional differences were the other risk factor for *Fasciola* spp. infection.

**Abstract:**

The objective of this investigation was to understand the epidemiology of fascioliasis in yaks in the alpine pastoral areas of the Qinghai–Tibet Plateau, China. The prevalence of *Fasciola hepatica* infection was estimated by examining eggs in the feces of yaks and by autopsy after the slaughter. Yaks were sampled from a total of 16 representative counties in Qinghai province, and risk factors were assessed based on regional and age characteristics. Fecal samples were obtained from 1542 yaks aged 0–1 (<1 year old), 1–2 (≥1 year old and <3 years old), and over 3 years (≥3 years old). In addition, 242 yaks over 3 years old who had not undergone fecal examinations were randomly selected for autopsy. A total of 267 fecal samples were positive for *Fasciola* spp. eggs. The average infection rate was 17.32% (0–60.61%), and the average infection intensity was 51.9 eggs per gram (epg) of feces, with intensities ranging from 18 to 112 epg. In Maduo, Dari, Zhiduo, Chengduo, and Datong counties, the *Fasciola* spp. eggs infection rate was zero. *Fasciola* spp. adult flukes were detected in 66 out of 242 yaks at autopsy, with a total infection rate of 27.27% and an average infection intensity of 21.2 (adult worms), with intensities ranging from 3 to 46 worms. Logistic regression model analysis showed that age was a significant risk factor for yak infection with *Fasciola* spp. In addition, the risk varied between regions: Haiyan, Gangcha, Duran, and Wulan were all high-risk areas for yak infection with *Fasciola* spp. The spatial distribution of the *Fasciola* spp. infection rate in each region showed a very weak negative correlation (Moran’s I = −0.062), Duran formed a spatial distribution of high–low clusters with surrounding areas, and Datong formed a low–high clustering distribution characteristic with the surrounding areas. This investigation revealed that the infection rate of *Fasciola* spp. in yaks was higher on the Qinghai–Tibet Plateau. Increasing age was a risk factor for infection with *Fasciola* spp.; different regions also have a different risk of *Fasciola* spp. infection. Only two regions showed clustering characteristics in the spatial distribution of infection rates. These findings extend the epidemiological information on *Fasciola* spp. infection in yaks and provide baseline data for the execution of control measures against *Fasciola* spp. infection.

## 1. Introduction

The yak is a unique cattle species produced by severe natural selection and self-adaptation in the Qinghai–Tibet Plateau and its adjacent areas. The yak is a unique biological resource of the plateau in China and cannot be replaced by other animal species. Yaks are mainly distributed on the Qinghai–Tibet Plateau and its adjacent areas, with a population of more than 14 million [1,2]. Yaks provide local herdsmen with production and daily necessities, such as milk, meat, wool, labor, and fuel, which are closely related to production and the daily lives of the local people [3].

Fascioliasis is an emerging neglected infection caused by *Fasciola* spp. that affects animals and human health and well-being [4]. *Fasciola* infection is caused by ingestion of encysted metacercariae; the infected hosts show acute symptoms (fever and abdominal pain) or chronic symptoms (intermittent pain, jaundice and anemia, pancreatitis, liver fibrosis or abscess or cirrhosis, cholangitis, cholecystitis, obstruction of bile flow, and gallstones), and even death [5,6,7,8,9,10,11]. Several studies have shown that *Fasciola* spp. is the dominant species of yak parasites [12,13,14,15,16]. After yaks are infected with *Fasciola* spp., their working ability is reduced, their growth is slowed, their development is blocked, production performance and commercial value decrease significantly, and large quantities of slaughtered yak livers that cannot be sold as livestock products are wasted. In particular, this can lead to the death of a large number of young yaks, causing great economic loss to the yak breeding industry while posing a threat to human health. 

Qinghai province of the Qinghai–Tibet Plateau is the source region of China’s Three Rivers and has a low air pressure, a low temperature, a large temperature difference between day and night, little but concentrated rainfall, a long period of sunshine, strong solar radiation, and a low oxygen content [17,18]. Winter is cold and long, while summer is cool and short. There are clear differences in climate among the different regions. The altitude is above 3000 m, and there are many rivers, streams, and marshes in most regions, which is suitable for the survival and reproduction of the intermediate hosts of *Fasciola* spp. Yaks rely on natural pastures for feeding. Due to the lack of infrastructure, herdsmen and yaks drink water from nearby rivers and are vulnerable to parasitic diseases with a variety of pathogens, including *Fasciola* spp. This phenomenon becomes a favorable condition for the life cycle of *Fasciola* spp.

*Fasciola* spp. are the major species of yak parasites, presenting a regional epidemic fascioliasis; however, information on the epidemiology of *Fasciola* spp. infection in yaks in the pastoral areas of Qinghai province is limited. The objective of this study was to investigate the prevalence of yak fascioliasis in pastoral areas of the Qinghai Plateau through fecal and necropsy examinations, aiming to provide baseline data for the prevention and control of yak fascioliasis in China.

## 2. Materials and Methods

### 2.1. Origin of Experimental Animals

From March to April 2020, 1784 yaks were selected as a random subset from yak farms in 16 counties in Qinghai Province, comprising 1542 yaks for fecal tests and 242 yaks for autopsy. All yaks were over 3 years old at autopsy, and the detailed sample collection information is shown in Table 1.

### 2.2. Examination of Fasciola spp. Eggs in Feces

The random sampling method was adopted. Fresh fecal samples of the yaks were collected through the rectum before grazing in the morning; they were numbered, and their genders, ages, and geographic locations were recorded. The samples collected (each sample was ≥50 g) were placed in zip-locked bags, placed into refrigerated boxes, taken back to the laboratory, and placed in a 4 °C refrigerator until examination.

The fecal sedimentation technique was used for examining *Fasciola* spp. eggs; the number of eggs observed was counted under a microscope, and the number of eggs per gram (epg) of feces was calculated [19,20,21].

### 2.3. Examination and Identification of Adult Worms

We randomly selected 242 yaks over 3 years old without fecal examination from 16 different localities to be autopsied. These autopsy samples were all collected from 16 different regional slaughterhouses; they are shown in Table 2. The flukes recovered from the liver and cholecyst were washed immediately in saline and preserved in 70% ethanol. The identification of *Fasciola* spp. was based on morphological characteristics [19].

### 2.4. Molecular Identification of Adult Worms

According to the published sequences of the internal transcribed spacers (ITS) of *F. hepatica* that are available in the GenBank (accession number: AJ628432, AJ628431), primers for the amplification of ITS-1 and ITS-2 of *F. hepatica* were designed: ITS-1 primers: F1: CTCATTGAGGTCACAGCAT, F2: CAATGGCAAAGAATGGCAAG; ITS-2 primers: F3: ATATTGCGGCCATGGGTTAG, F4: CCAATGACAAAGTGACAGCG. PCR amplifications of the above regions were performed in 50 µL of a reaction mixture containing 22 L of 2×Taq Mastermix (Qiagen, Hilden, Germany), 0.5 µL of each primer (50 pmol/µL), and 4 µL of template DNA; DNase/RNase-free deionized water was added to bring the volume to 50 µL. A negative control (without DNA) was included in each PCR reaction. The PCR amplification conditions were as follows: initial denaturation at 98 °C for 5 min, followed by 30 cycles consisting of denaturation at 94 °C for 30 s, annealing at 52 °C for 30 s, and extension at 72 °C for 30 s. A final elongation step was conducted at 72 °C for 5 min at the end of the amplification procedure.

### 2.5. Prevalence of Fasciola spp.

The following calculations were made using the appropriate formulas: infection rate = (number of infected animals/number of examined animals) × 100%. Mean infection intensity = number of detected (eggs or worms)/number of infected animals examined. Range of infection = minimum positive infections − maximum positive infections. *Fasciola* spp. eggs were all counted as epg, the number of epg in the fecal sample. Subsequent representations of the egg infection intensity were used in the following format: epg of feces (minimum number of infected epg of feces − maximum number of infected epg of feces).

### 2.6. Statistical Analysis

The differences in prevalence between regions were analyzed using the Bonferroni chi-squared test, and the differences in prevalence between age groups were analyzed using the same method. Variables that were significant (*p* ≤ 0.1) at the 95% confidence level were tested for collinearity and selected for inclusion in the multivariate logistic regression models. The local spatial autocorrelation test was used to analyze the spatial clustering characteristics of infection rates in different regions, and the results are shown as Moran scatter plots and clustering maps. All data were analyzed using the SPSS statistical package (version 26), and the spatial characteristics data were analyzed using GeoDa software. All plots were generated using GraphPad Prism 9.0 and RStudio software (version 4.2.0).

## 3. Results

### 3.1. Identification of Eggs

Eggs of *Fasciola* spp. from yaks are broadly ellipsoidal, operculated, yellow–brown, measuring 136.43–155.14 µm long by 69.39–87.52 µm wide, and are shedded unembryonated in feces (Figure 1).

### 3.2. Morphological and Molecular Identification of Adult Worms

Fresh worms were fleshy red and became gray–white after fixation. The size of the worms was 20–75 mm × 8–20 mm. The collected fresh worms were fixed and shown below (Figure 2a), and an adult worm stained by German hematoxylin is shown below (Figure 2b). The sequences of PCR-amplified ITS-1 and ITS-2 were completely consistent with the corresponding *F. hepatica* sequences available in GenBank.

### 3.3. Results of the Fecal Examination

#### 3.3.1. Infection of *Fasciola* spp. by Fecal Examination

The results of fecal examination of *Fasciola* spp. eggs in yaks from different areas are shown in Table 3. A total of 1542 yak fecal samples were collected from 16 regions, and 267 were positive for *Fasciola* spp. eggs. The average infection rate was 17.32% (0–40.21%), and the average infection intensity was 51.9 epg (18–112).

#### 3.3.2. *Fasciola* spp. Infection in Yaks of Different Ages

Table 4 shows the intensity of *Fasciola* spp. infection in yaks of different ages. The infection rate was 9.90% (0–25.00%) in 0–1 year old yaks, 16.18% (0–37.50%) in 1–2 year old yaks, and 25.88% (0–60.61%) in yaks over 3 years old. A comparison of infection rates between field subgroups using a multigroup Bonferroni chi-squared test revealed significant differences in the infection rates of yaks among the three different age groups (*p* < 0.0167). The infection rate was significantly lower in the group of 0–1 year olds than in the other two age groups and significantly higher in the over 3 year olds group than in the other two age groups (Figure 3). The mean infection intensity was 51.9 epg (18–112), comprising 45.5 (18–92) in yaks aged 0–1 year, 52.0 (23–93) in yaks aged 1–2 years, and 58.1 (28–112) in yaks aged over 3 years. The results showed that the fecal infection rate and infection intensity of *Fasciola* spp. increased with the increase in yak age.

#### 3.3.3. Differences in *Fasciola* spp. Egg Infection in Yaks from Different Areas

The areas with positive yak fecal samples infected with *Fasciola* spp. are shown in Figure 4a. A comparison of infection rates across site subgroups using a multigroup Bonferroni chi-squared test revealed that Gonghe and Qilian counties had significantly lower infection rates than the other sampling sites, while Wulan had a significantly higher infection rate than the other sites, *p* < 0.001 (Figure 4b).

#### 3.3.4. Spatial Characteristics of *Fasciola* spp. Infection in Yaks

The geographical distribution of *Fasciola* spp. infection in yaks is shown in Figure 5a. The spatial correlation of *Fasciola* spp. infection rates was explored based on yak infection rates in different regions as a single variable to derive correlations of yak infection rates in different regions. Spatial autocorrelation tests yielded a Moran index of −0.062 and Moran scatter plots distributed in quadrants two and four, indicating that yak infection rates showed a negative spatial correlation (Figure 5b). Regions with high infection rates showed mutual exclusion from other areas with high infection rates, tending to be closer to areas with low infection rates. The significantly spatially linked areas are Datong (*p* < 0.05) and Duran (*p* < 0.05) (Figure 5c); Duran formed a spatial distribution of high–low clusters with surrounding areas, and Datong formed a low–high clustering distribution characteristic with the surrounding areas (Figure 5d).

### 3.4. Autopsy Results

The autopsy results of the yak *Fasciola* spp. infection are shown in Table 5. A total of 242 yaks were examined, comprising 210 slaughtered yaks and 32 dissected yaks. A total of 66 yaks were found to be infected with *Fasciola* spp. The average infection rate was 27.27% (0–39.13%), and the average infection intensity was 21.2 worms (3–46). 

### 3.5. Assessment of Risk Factors for Fasciola spp. Infection of Yaks Based on Areas and Age Groups

Logistic regression analysis of high-risk factors for *Fasciola* spp. infection in yaks was carried out using the presence of *Fasciola* spp. infection in yaks as the dependent variable and the age group and region as the independent variables. The results showed that both factors are included in the logistic model. In particular, the 2 year old group showed an increased risk of being infected with *Fasciola* spp. (odds ratio (OR) = 2.816; *p* < 0.01), and the risk was higher for the over 3 year old group (OR = 3.921; *p* < 0.001). Being from Haiyan (OR = 2.334; *p* < 0.05) increased the risk of *Fasciola* spp. infection in yaks, while yaks from Gangcha (OR = 2.985; *p* = 0.002), Duran (OR = 3.130; *p* = 0.001), and Wulan (OR = 3.348; *p* = 0.001) had a high risk of *Fasciola* spp. infection (Table 6). An optimized multiple logistic regression of the two factors, age and area, was conducted to obtain a risk assessment model. It was found that Qilan had the lowest risk score among the areas with *Fasciola* spp. infection, while Wulan had the highest risk score, with the lowest score for the 1 year old age group and the highest score for the over 3 year old age group among the different age groups. The two-factor combination with the highest risk score was the over 3 year old age group, with a combined total score of over 180 for Wulan, which corresponds to a risk of disease greater than 0.5, while the 1 year old age group achieved a combined total score close to 0 in Qilian, which corresponds to a risk of disease of less than 0.1, the two-factor combination with the lowest risk score (Figure 6).

## 4. Discussion

Of the 1542 tested yak fecal samples from 16 regions, 267 (17.32%) were positive for *Fasciola* spp. with a mean intensity of 51.9 epg, and the regional prevalence ranged from 0% (Maduo, Dari, Zhiduo, Chengduo, Zeku, and the Datong yak farm) to 40.21% (Wulan). Among different age groups of yaks, the prevalence of *Fasciola* spp. ranged from 9.90% to 25.88%. By autopsy, the prevalence of *Fasciola* spp. infection in yaks was 27.27% (66/242) with a mean intensity of 21.16 worms. To the best of our knowledge, this is the first investigation of *Fasciola* spp. infection in yaks in Guinan, Gangcha, Dulan, Wulan, Dari, Maduo, Zhiduo, Chengduo, Henan, Zeku, and Datong counties in Qinghai province, China. In the previous traditional studies, most researchers showed that yaks were infected with *F. hepatica*. During 1996 to 2020, the prevalence of *F. hepatica* was reported in major yak-producing areas in China such as the Tibetan Autonomous Region (18.33–100%) [16,22,23], some counties in Qinghai province (16.7–97%) [24,25,26,27,28,29], the northwest of Sichuan province (23%) [30], and Gansu province (6.55–42.5%) [15,31]. In addition, the prevalence of Fasciola was 10% in India and 16.67% in Nepal [32,33]. These studies indicated that *F. hepatica* infection is common in yaks in the Qinghai–Tibet Plateau regions, although the prevalence varied across different geographical localities.

Recently, Kong Xiangying et al. [34] found the first *Fasciola gigantica* in yaks in Haibei, Qinghai Province. Gao Xing et al. [35] isolated the *Fasciola intermediate* and characterized the mitochondrial genome in yaks from Haibei, Qinghai province. In this study, we identified the morphology of the adult worms and selected some adult samples for molecular biological identification. The PCR amplification and sequencing results of these samples were consistent with the sequences of *F. hepatica*, which did not exclude the possibility of *F. gigantica* and *F. intermediate* in the remaining samples.

In the present study, the location and age of yaks were associated with *Fasciola* spp. infection in yaks in the investigated areas. The trend of *Fasciola* spp. prevalence increased with the age of yaks, which was in accordance with previous reports [15,16]. Under the same grazing mode, the difference in *Fasciola* spp. prevalence may be caused by a difference in immunity, feeding conditions, management measures (e.g., prevention and control measures; degrees of repeated grazing; and types, frequency of use, and dosage of anthelmintic), regional ecological environment (e.g., number of rivers, mountains, low-lying swamps, altitude, moisture, temperature, and *Lymnaea stagnalis*), and sampling season. For example, in the survey areas with a high positive rate, there were more wetlands and grasslands that closely border marshes; these areas have similar water environmental conditions and many wild animals that drink water from the same nearby rivers that domestic animals drink from. The infected yaks could shed *Fasciola* spp. eggs into the environment. The intermediate host of *Fasciola* spp. is *L. stagnalis*, which becomes infected by swallowing the *Fasciola* spp. eggs excreted by yaks. Then, the infected *L. Stagnalis* sheds cercariae into the environment from July to October. The cercariae then develop into cysticerci and infect hosts, such as yaks and sheep, in rivers and grasslands. This creates a vicious cycle of persistent *Fasciola* spp. infection. It is interesting to note that the yaks sampled at the Datong yak farm had no *Fasciola* spp. infection. This is because the Datong yak farm is located in the interlacing zone of agriculture and pasture, the disinfection procedures are more standardized, and the deworming frequency is more intensive; thus, *Fasciola* spp. infection is better controlled.

In addition, the current study found that the infection rate of *Fasciola* spp. in yaks did not show obvious spatial characteristics, and the regions occasionally show a pattern of low–high clustering and high–low clustering. The regions showing this local clustering are Datong and Duran counties. It is speculated that the reason for this cluster pattern is the difference in the natural geography and grazing characteristics between the two places and the surrounding bordering areas. In other words, the disease flow factors in the regions and the surrounding areas are not continuous, and the epidemic factors are discontinuous in the geographical pattern. The distribution of disease epidemic factors tends to be discrete.

## 5. Conclusions

This survey showed a high *Fasciola* spp. prevalence (17.32% by fecal test and 27.27% by autopsy) in yaks in 16 counties of Qinghai province in the Qinghai–Tibet Plateau, which causes economic losses to the local yak industry and poses a potential threat to the health of the native population. Moreover, the location and age of yaks in the surveyed area are related to the *Fasciola* spp. infection. These findings not only extend the epidemiological information on *Fasciola* spp. infection in yaks but also provide useful baseline data for the prevention and control of fascioliasis in yaks.

## Figures and Tables

**Figure 1 animals-13-03330-f001:**
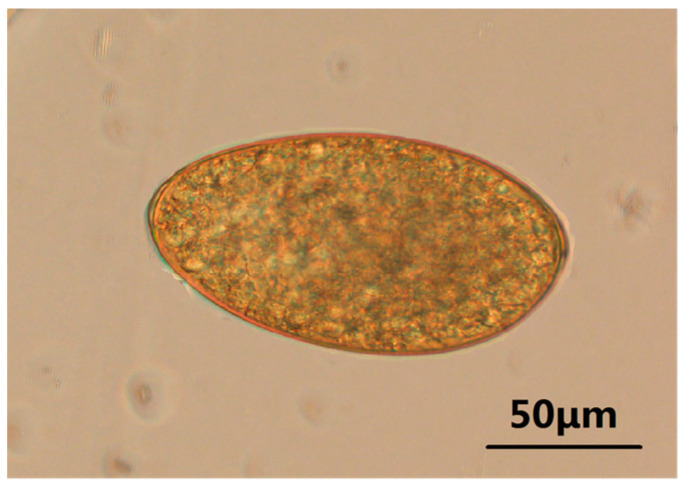
Identification of *Fasciola* spp. eggs isolated from feces. The complete *Fasciola* spp. egg form is visible, golden-yellow, long elliptic or oval, narrow at one end, and blunt at the other end (multiple: 10 × 10).

**Figure 2 animals-13-03330-f002:**
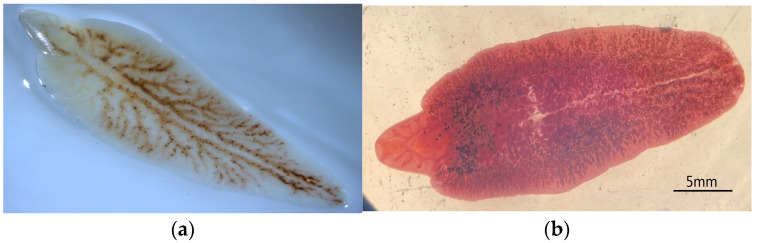
Morphology of *Fasciola* spp.; (**a**) sample of collected worm; and (**b**) an adult worm stained with German hematoxylin.

**Figure 3 animals-13-03330-f003:**
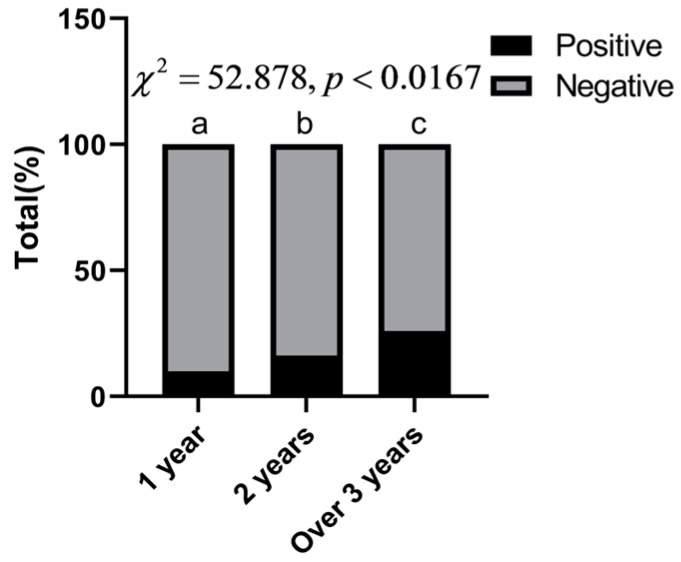
Bar chart and differences in infection rate in yaks by age group (groups marked with the same letter are not significantly different).

**Figure 4 animals-13-03330-f004:**
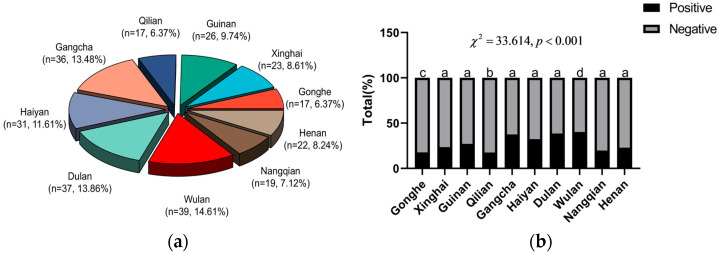
Examination results of *Fasciola* spp. infection in yaks from different areas; (**a**) regional composition of positive samples; and (**b**) group bar charts of regional infection conditions (groups marked with the same letter are not significantly different).

**Figure 5 animals-13-03330-f005:**
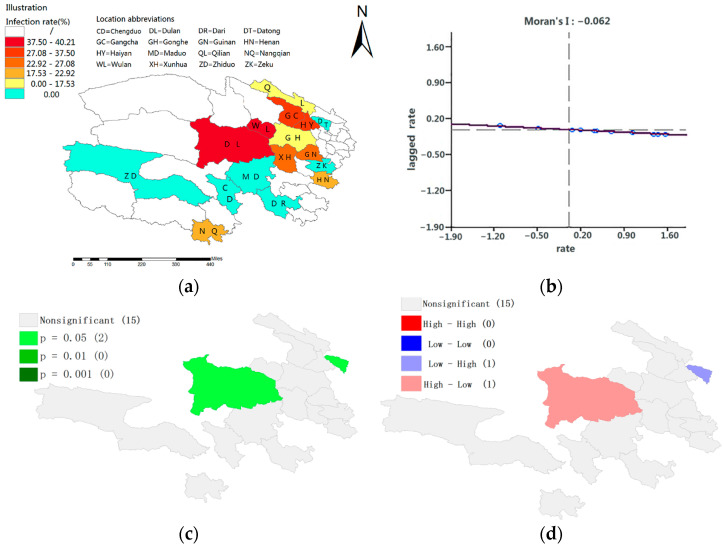
Examination results of *Fasciola* spp. infection rate in yaks in different areas spatial features: (**a**) map of the regional distribution of *Fasciola* spp. infection rates in yaks; (**b**) univariate Moran scatter plots of *Fasciola* spp. infection in yaks in different regions; (**c**) spatially relevant areas with significance; and (**d**) areas where clustering characteristics of infection rates are spatially significantly correlated.

**Figure 6 animals-13-03330-f006:**
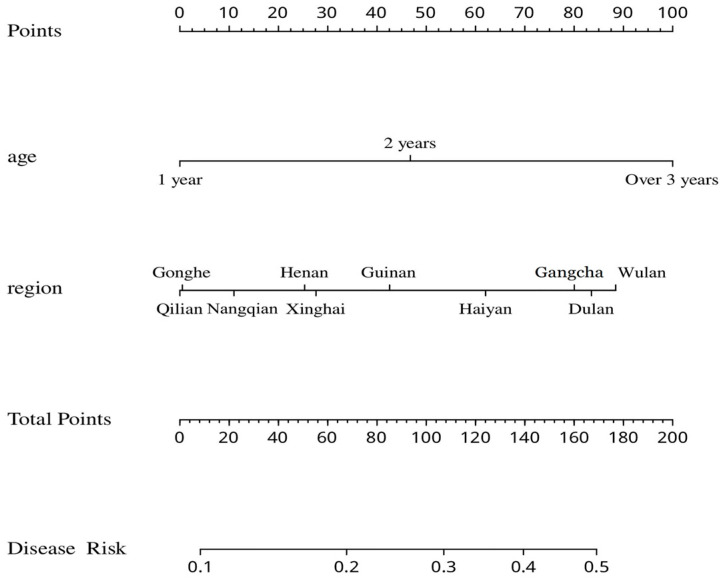
A disease risk assessment graph (nomogram) generated using a logistic regression model based on sample regions and yak age. The positions of the scales for different ages and regions correspond to the first row. For each factor, one can take the score for the corresponding position in the top row, add up the scores for all factors, find the scale for the corresponding total score in the fourth row, and then observe the risk of disease below the total points relative position.

**Table 1 animals-13-03330-t001:** The information about the samples collected in this study.

Areas	Geographic Coordinates	Altitude (m)	Annual Precipitation (mm)	Fecal Test (Number)	Autopsy (Number)
Gonghe	E 100°62′66.23″, N 36°28′87.03″	above 3200	371.0	96	22
Xinghai	E 99°58′57.18″, N 35°35′18.25″	above 3306	360.0	98	21
Guinan	E 100°13′–101°31′, N 35°09′–36°08′	about 3100	488.0	96	22
Qilian	E 100°26′40.41″, N 38°03′44.15″	above 2717	406.7	97	26
Gangcha	E 100°08′45.00″, N 37°19′31.69″	above 3700	345.0	96	23
Haiyan	E 100°59′39.95″, N 36°53′40.81″	above 3008	447.4	96	22
Dulan	E 98°06′09.74″, N 36°18′27.40″	above 3041	213.4	96	24
Wulan	E 98°48′67.39″, N 36°93′57.48″	above 3100	210.4	97	26
Dari	E 99°39′3.99″, N 33°45′12.55″	above 3900	531.5	96	20
Maduo	E 97°19′15.99″, N 34°57′10.55″	above 4290	585.5	96	20
Yushu	E 97°00′87.85″, N 32°99′31.07″	above 3681	487.0	97	22
Chengduo	E 97°06′39.00″, N 33°22′09.18″	above 3851	500.0	96	20
Nangqian	E 96°48′58″, N31°32′20″	above 3650	527.3	96	22
Henan	E 102°08′56.36″, N 34°29′16.96″	above 3520	597.1	96	20
Zeku	E 100°34′–102°08′, N 34°45′–35°32′	above 3650	564.9	96	20
Datong	E 101°38′00.28″, N 37°25′23.12″	above 2950	547.2	97	32
Total				1542	242

**Table 2 animals-13-03330-t002:** The slaughterhouse source of the autopsy samples.

Number	Slaughterhouse
1	Qinghai Lake meat industry Co., Ltd., slaughterhouse (Gonghe, China)
2	Xinghai County green grass source food Co., Ltd., slaughterhouse (Xinghai, China)
3	Guinan County Lvjiayuan cattle and sheep slaughterhouse (Guinan, China)
4	Qilian Yida animal products Co., Ltd., slaughterhouse (Qilian, China)
5	Gangcha County Yipin animal products Co., Ltd., slaughterhouse (Gangcha, China)
6	Haiyan County Huaxia cattle and sheep slaughterhouse (Haiyan, China)
7	Qinghai Kaitai agriculture and animal husbandry Co., Ltd., slaughterhouse (Dulan, China)
8	Wulan County Hengcheng beef and mutton slaughterhouse (Wulan, China)
9	Guoluo Haoyun designated Cattle and Sheep Co., Ltd. (Dari, China)
10	Guoluo Jin Grassland yak slaughtering and processing Co., Ltd. (Maduo, China)
11	Zhiduo County meat food Co., Ltd., slaughterhouse (Zhiduo, China)
12	Chengduo County plateau yak livestock products Co., Ltd., slaughterhouse (Chengduo, China)
13	Yushu Muyuan meat industry Co., Ltd., slaughterhouse (Yushu, China)
14	Sanjiang Ranch Co., Ltd., slaughterhouse (Henan, China)
15	Qinghai Northwest Hong organic resources development Co., Ltd., cattle and sheep slaughterhouse (Zeku, China)
16	Qinghai Datong Jinlu Industry and Trade Co., Ltd., slaughterhouse (Datong, China)

**Table 3 animals-13-03330-t003:** The fecal worm egg infection of 1542 yaks *Fasciola* spp. in different areas.

Areas	Number of Investigated Yaks	Positive Rate (%)	Mean Intensity (epg)
Gonghe	96	17.71 (17/96)	53.3 (29–79)
Xinghai	98	23.47 (23/98)	46.3 (21–66)
Guinan	96	27.08 (26/96)	45.6 (20–70)
Qilian	97	17.53 (17/97)	49.7 (23–76)
Gangcha	96	37.50 (36/96)	57.9 (30–112)
Haiyan	96	32.29 (31/96)	53.8 (28–97)
Dulan	96	38.54 (37/96)	55.7 (27–82)
Wulan	97	40.21 (39/97)	63.1 (36–103)
Dari	96	0	0
Maduo	96	0	0
Zhiduo	97	0	0
Chengduo	96	0	0
Nangqian	96	19.79 (19/96)	45.1 (18–84)
Henan	96	22.92 (22/96)	48.1 (32–59)
Zeku	96	0	0
Datong	97	0	0
Total	1542	17.32 (267/1542)	51.9 (18–112)

**Table 4 animals-13-03330-t004:** The intensity and rate of *Fasciola* spp. infection in yaks of different ages.

Areas	0–1 (<1 Year Old)		1–2 (≥1 Year Old and <3 Years Old)		Over 3 Years (≥3 Years Old)	
	Number	Mean Intensity (Epg)	Number	Mean Intensity (Epg)	Number	Mean Intensity (Epg)
Gonghe	32	47.4 (29–62)	32	53.2 (37–64)	30	59.3 (34–79)
Xinghai	33	42.3 (21–58)	32	46.6 (30–59)	31	50.1 (28–66)
Guinan	32	39.5 (20–62)	32	44.8 (27–70)	31	52.6 (36–68)
Qilian	32	43.9 (23–53)	32	50.3 (33–68)	30	54.8 (33–76)
Gangcha	33	50.2 (30–92)	32	57.4 (38–89)	31	66.1 (29–112)
Haiyan	33	45.7 (28–71)	32	53.0 (39–82)	32	62.6 (40–97)
Dulan	32	48.9 (27–54)	32	57.7 (31–69)	32	60.4 (39–82)
Wulan	34	55.1 (36–79)	32	63.2 (39–93)	32	71.0 (43–103)
Dari	32	0	32	0	32	0
Maduo	32	0	32	0	32	0
Zhiduo	33	0	32	0	32	0
Chengduo	32	0	32	0	32	0
Nangqian	32	38.7 (18–54)	32	45.5 (25–61)	31	51.2 (37–84)
Henan	32	43.5 (32–54)	32	48.5 (41–58)	32	52.7 (43–65)
Zeku	0	0	0	0	0	0
Datong	33	0	33	0	32	0
Total	515	45.5 (18–92)	513	52.0 (23–93)	514	58.1 (28–112)
Infection rate	9.90%		16.18%		25.88%	

**Table 5 animals-13-03330-t005:** Investigation of *Fasciola* spp. infection in yaks in different areas by autopsy.

Study Areas	Number of Yaks Examined	Prevalence (%)	Mean Intensity (Number of Worms)
Gonghe	22	22.73 (5/22)	9.2 (3–22)
Xinghai	21	28.57 (6/21)	17.5 (4–37)
Guinan	22	27.27 (6/22)	14.2 (6–26)
Qilian	26	19.23 (5/26)	11.6 (6–18)
Gangcha	23	39.13 (9/23)	30.6 (7–46)
Haiyan	22	31.82 (7/22)	22.7 (3–21)
Dulan	24	37.50 (9/24)	28.3 (4–34)
Wulan	26	38.46 (10/26)	28.6 (9–39)
Dari	20	0	0
Maduo	20	0	0
Yushu	22	0	0
Chengduo	20	0	0
Nangqian	22	22.73 (5/22)	20.1 (7–43)
Henan	20	20.00 (4/20)	16.8 (12–36)
Zeku	20	0	0
Datong	32	0	0
Total	242	27.27 (66/242)	21.2 (3–46)

**Table 6 animals-13-03330-t006:** Risk factors for *Fasciola* spp. infection in yaks.

Factor	Group	n	Prevalence (%)	OR	95% CI for the OR		*p*-Value
					Upper Limit	Lower Limit	
Age	1 year	322	15.84	1.000			
	2 years	320	25.94	1.895	1.275	2.816	0.002
	Over 3 years	322	41.30	3.921	2.681	5.736	<0.001
Areas	Qilian	97	17.53	1.000			
	Gonghe	96	17.71	1.007	0.472	2.146	0.986
	Xinghai	98	23.47	1.459	0.711	2.993	0.303
	Guinan	96	27.08	1.789	0.881	3.632	0.108
	Gangcha	96	37.50	2.985	1.503	5.927	0.002
	Haiyan	96	32.29	2.334	1.166	4.675	0.017
	Dulan	96	38.54	3.130	1.578	6.207	0.001
	Wulan	97	40.21	3.348	1.693	6.619	0.001
	Nangqian	96	19.79	1.162	0.553	2.440	0.692
	Henan	96	22.92	1.413	0.685	2.917	0.349
Total		964	27.70				

The results of the analysis are presented as the number of yaks (n), odds ratio (OR), and 95% confidence intervals (CI).

## Data Availability

Not applicable.

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
