# Peer review of "Epidemiological Investigation of Yak (Bos grunniens) Fascioliasis in the Pastoral Area of Qinghai–Tibet Plateau, China"

_animals, 2023, doi:10.3390/ani13213330_

Round 1
Reviewer 1 Report (Previous Reviewer 1)
The manuscript is better now, but there should be some more changes.
L 186: “F. hepatica” italics
Table 4: “Wulan 40.6 (9-39)” I think there is a mistake. Also, correct the alignment.
Table 5: 1st lines of the factors “age” and “areas” do not have 95% CI for the OR and P-value
L 255, 257, 263: “F. hepatica” italics
L 258-261: The references are not presented correctly
L281: “In is interesting” “It is interesting”
Author Response
On behalf of all co-authors, I would like to thank you and the reviewers very much for further positive comments and constructive suggestions on our manuscript (MS) ID animals-2396635.R1. We have revised the MS strictly according to the reviewers’ comments and suggestions. Please see the attachment.

Reviewer 2 Report (Previous Reviewer 4)
The resubmitted manuscript has been revised following reviewer comments. It can be accepted for publication
Author Response
On behalf of all co-authors, I would like to thank you and the reviewers very much for further positive comments and constructive suggestions on our manuscript (MS) ID animals-2396635.R1. We have revised the MS strictly according to the reviewers’ comments and suggestions. Please see the attachment.

Reviewer 3 Report (New Reviewer)
The authors present a study about the prevalence of Fasciola in yaks in the Tibet Plateau, China. The work is interesting from an epidemiological point of view to shed light on the distribution of this helminth in that area. However there are some major issues in this work. The whole work is assuming the eggs and adult worms analyzed are F. hepatica. However, there is no enough evidence to support this statement. Actually, the fluke showed in Figure 2 is more similiar to F. gigantica (less shoulders, elongated) than to F. hepatica. Authors mention in the methods performing a PCR to confirm the species but they never mentioned the results.
A paper published last year about a Fasciola phylogenetic analysis from infected yaks in the same area (PMID: 35647106) reveal the form of intermediates forms (F.hepatica-F. gigantica) so it would be a big mistake to consider the parasite in the present work F. hepatica.
I would recommend the authors to rewrite the manuscript substituting F. hepatica for Fasciola sp. and add a paragraph in the introduction and in the discussion about Fasciola intermediate forms.
Besides this observation, other issues should be addressed:
- Correct Fasciola throughout the document to be in italics.
- Avoid repetition of "in yaks" in the simple summary and abstract.
- Line 59. Remove through the oral route. Ingestion already implies oral route.
- Line 60. Revise. Patients can have acute and chronic symptons, they are not exclusive one from another.
- Line 100. Why didn´t you use kato katz technique? It is the gold standard for epg quantification.
- Line 105. Add from slaughterhouse.
- Table 2. Number of yaks analyzed (heading)
- Add the age range of the yaks. Why did you use that age distinction? Why not 1-3 years, 5-10 and >10 years old?
- Table 3. It has a star underneath the table but it is not refered to anything in the table.
- Figure 4 does not provide any valuable information since the infection rate is already included in the previous table.
- Table 4. Center first line of data.
- Explain nomogram better in the text.
- Lines 258, 259, 260. References error.
- L. stagnalis in italics.
English language should be revised:
- Revise second sented of simple summary, doesn´t make sense.
- Line 143. Replace passed from shed.
- Line 276- Replace discharge by shedding.
Author Response
On behalf of all co-authors, I would like to thank you and the reviewers very much for further positive comments and constructive suggestions on our manuscript (MS) ID animals-2396635.R1. We have revised the MS strictly according to the reviewers’ comments and suggestions. Please see the attachment.

This manuscript is a resubmission of an earlier submission. The following is a list of the peer review reports and author responses from that submission.
Round 1
Reviewer 1 Report
The article is very interesting. There have to be some minor changes.
L20: “of Qinghai” delete the extra space
L29: “51.9 per gram of faeces (epg),”rewrite it “51.9 eggs per gram (epg) of faeces,”
L52: “is an and emerging” delete “and”
L57: “. Patients” not “, Patients”
L7-80: “This phenomenon becomes a 79 favorable condition for the life history cycle of F. hepatica.” Please add a reference.
L84: “Qinghai Province” should be “Qinghai province”
L93: “autopsy. and” should be “autopsy and”
Table 1: “Number (fecal test) and Number ( Autopsy)” should be “fecal test (Number) Autopsy (Number)”
L113-114: “The amount of feces in the human centrifuge tube is 5 mL” why do you mention human?
L114: “,and” add space
L115: “fluke fluke” is double
L115- 116: “eggs (EPG) per gram of feces” should be “eggs per gram (epg) of feces”
L121, L141, L142, L186, L197: “worms(adult)” add space
L151, L153, L154: “98 ℃, 52 ℃, 72 ℃, 72 ℃” delete the spaces
L152: “94 ℃for” delete space, add space
L160, L162, L163: “eggs per gram” should be “epg”
L162: “faeces(minimum” add space
L181- 182: “Here is the clearly visible F. hepatica egg with a long axis of 136.43 μm we caught” rewrite it more formal style
L198: “low(Figure” add space
L198-200: “PCR amplification of T. sinensis was performed with specific ITS-1 and 198 ITS-2 primers used to amplify specific bands in both ITS-1 and ITS-2, and the sequencing 199 results were completely consistent with the sequence alignment of F. hepatica ITS.” Why do you refer to T. sinensis?
L201: “hepatica. (a)” and “worm. (b)” not “.” But “; or ,”
L209: “cattle” should be “yaks”
L216: “F. hepatica” should be in italics
L224: “Table 3 shows the infection intensity of F. hepatica in yaks at different ages.” You repeat it in the begging of the paragraph.
Table 2, 3: I can’t understand where you are referring using “scope” and “head” and you do not explain it in the text. There should be, at least, a footnote.
L232, L241: “F. hepatica” italics
L233-234: “(Figure 5a)” without “()”
L238: “hepatica. (a)” and “samples. (b)”. no “.” But “; or ,”
Table 4: I can’t understand where you are referring using “strip” and “head” and you do not explain it in the text. There should be, at least, a footnote.
“areas” is better to be under “Investigation”
Table 5: “areas” and “total” in bold
In the footnote, you should explain “n”, as well.
Figure 7: The figure is very helpful, but it is a little blurred. Is there any possibility to be clearer?
L319: “cattle” should be “yak”
L355: there is reference that has not inserted
L339- 395: these paragraphs must be rewritten, as it is very tiresome to read the names of all the researchers. This is why we include the references.
Reviewer 2 Report
The study was to investigate the epidemiology of fasciolosis in yak in the alpine pastoral areas of Qing-hai-Tibet Plateau, China. There is limited information on the prevalence of fasciolosis in yaks. The epidemiological investigation of yak liver fasciolosis is lack of systematic research in the pastoral areas of Qinghai Province where yaks are widely distributed. The study will provide the basis for the study of the epidemiology and control of yak fasciolosis in this area. The MS is relatively well written, scientifically sound and the outcomes are relatively well explained, and it is of interest to the readership of Animals. Therefore, I recommend its acceptance for publication in Animals after minor revision according to the following comments.
Line 52 delete and.
Line 70 delete The.
Line 77 Fasciola hepatica should be F. hepatica.
Line 96 The title of 2.2 should be redefined.
Line 144 the software in the line should provide version.
There are minor grammatical mistakes and typo-errors throughout the MS, and should be corrected during revision.
Reviewer 3 Report
This manuscript describes coprological prevalence of Fasciola hepatica in yaks in Qinghai-Tibet Plateau, China. Although would be of interest to have epidemiological data from this area, I must say that manuscript does not meet criteria for scientific articles. First of all, it lacks originality. There are many articles on epidemiology of F. hepatica in cattle from 1960-2020. This MS does not bring anything new – no original diagnosis, design, no surprising epidemiologic data. The description of methods flaw at many places in the text. The fact that some regions are more abundant for liver flukes than others is very common. Unfortunately, authors did not explain the differences using spatial analysis in proper way. Discussion is difficult to follow and to get any output from the study. For instance, at lines 312-321 authors discuss role of intermediate snails that were not part of the study. Authors should be aware that their risk analyses did not include rainfall, pasture management or any other environmental factor. Without this, spatial data are useless.
quality of English is poor
Reviewer 4 Report
The manuscript describes a survey of Fasciola hepatica in the yak from the pastoral area of Qinhai-Tibet Plateau in China.
Although the number of the examined fecal samples and of the yaks examined after necropsy is impressive, the data is very badly presented and confusing.
The English language is very poor and needs a carful revision, possibly by an English speaking person. The manuscript is too long and boring. There are many repetitions that have to be removed. E.g. Line 56: there is no sense to repeat that it is a zoonotic disease because in the previous line it is state that it affects human heath also. I add some more examples, but it is impossible to list all the points that have to be corrected.
The authors use indifferently the term rate and prevalence although most of presented data are of prevalence. Nevertheless, the statistical analysis is well done and acceptable.
The description of the method of fecal examination is redundant as it is the same used in most parasitological laboratories. It should be removed or strongly reduced. The same for the recovery of adult worms and both the figures of the egg and adult parasites should be removed as well know to readers.
The conclusions practically repeat the results and they have to be rewritten highlighting the most significant results and hypnotizing some possible solution, even partial, to the current situation.
In conclusion, the manuscript needs a major revision. It should rewritten in many parts, shorten, the repetitions removed, and the English needs a deep revision.
Here some minor concerns as an example
Frequently the authors use “on” instead of “of”
Line 27: were examined, not investigated
Line 28: 267 positive eggs, rephrase: 267 fecal sample wre positive for F. hepatica eggs
Line 85: liver fascioliasis, fascioliasis is enough
Line 102: sedimentation instead of precipitation
Line 178: what does it mean T. sinensis
Line 409: it is not clear why grazing pasture resources are limited
Table 2 and 3: what do you mean with intensity scope
The English quality is very poor
